# Astrin-SKAP complex reconstitution reveals its kinetochore interaction with microtubule-bound Ndc80

**David M Kern[1,2], Julie K Monda[1,2†], Kuan-Chung Su[1†], Elizabeth M Wilson-Kubalek[3], Iain M Cheeseman[1,2]***

[1]Whitehead Institute for Biomedical Research, Cambridge, United States; [2]Department of Biology, Massachusetts Institute of Technology, Cambridge, United States; [3]Department of Cell Biology, The Scripps Research Institute, La Jolla, United States

**Abstract** Chromosome segregation requires robust interactions between the macromolecular kinetochore structure and dynamic microtubule polymers. A key outstanding question is how kinetochore-microtubule attachments are modulated to ensure that bi-oriented attachments are selectively stabilized and maintained. The Astrin-SKAP complex localizes preferentially to properly bi-oriented sister kinetochores, representing the final outer kinetochore component recruited prior to anaphase onset. Here, we reconstitute the 4-subunit Astrin-SKAP complex, including a novel MYCBP subunit. Our work demonstrates that the Astrin-SKAP complex contains separable kinetochore localization and microtubule binding domains. In addition, through cross-linking analysis in human cells and biochemical reconstitution, we show that the Astrin-SKAP complex binds synergistically to microtubules with the Ndc80 complex to form an integrated interface. We propose a model in which the Astrin-SKAP complex acts together with the Ndc80 complex to stabilize correctly formed kinetochore-microtubule interactions.

DOI: https://doi.org/10.7554/eLife.26866.001

**\*For correspondence:**
icheese@wi.mit.edu

†These authors contributed equally to this work

**Competing interests:** The authors declare that no competing interests exist.

## Introduction

The macromolecular kinetochore complex links chromosomes to dynamic microtubule polymers and harnesses the forces generated by microtubule growth and depolymerization to facilitate accurate chromosome segregation. The proteins that comprise this critical interface have been the subject of intense study in multiple organisms (*Cheeseman, 2014*). Many kinetochore components, including the core microtubule-binding Ndc80 complex, are conserved throughout eukaryotes (*Cheeseman and Desai, 2008*). The Ndc80 complex associates with microtubules using a precise binding site and angled orientation (*Alushin et al., 2010*; *Wilson-Kubalek et al., 2016*). In addition to the Ndc80 complex, other kinetochore-localized microtubule binding proteins have been identified that act together with the Ndc80 complex to stabilize microtubule attachments or facilitate processive interactions at the outer kinetochore. In budding yeast, the Dam1 complex provides a ring-like coupler to stably attach each kinetochore to a depolymerizing microtubule by conferring processivity to the Ndc80 complex (*Lampert et al., 2010*). In metazoans, the Ska1 complex adds additional microtubule binding and confers its microtubule-tracking ability to the Ndc80 complex (*Schmidt et al., 2012*; *Welburn et al., 2009*). In this study, we implicate the vertebrate-specific Astrin-SKAP complex as an important additional player in forming an integrated kinetochore-microtubule interface.

We and others previously isolated a complex of Astrin, SKAP, and the dynein light-chain LC8 from human cells that localizes to both kinetochores and spindle microtubules (*Dunsch et al., 2011*;

*Schmidt et al., 2010*). Astrin and SKAP play critical roles in chromosome segregation and the maintenance of spindle bipolarity based on depletion and knockout experiments in human cells (*Dunsch et al., 2011*; *Friese et al., 2016*; *Gruber et al., 2002*; *Kern et al., 2016*; *Mack and Compton, 2001*; *Manning et al., 2010*; *McKinley and Cheeseman, 2017*; *Schmidt et al., 2010*; *Thein et al., 2007*). Within the Astrin-SKAP complex, the SKAP subunit contains two important microtubule-binding activities. First, SKAP possesses a microtubule plus-end tracking activity through its interaction with EB family proteins (*Friese et al., 2016*; *Kern et al., 2016*; *Tamura et al., 2015*; *Wang et al., 2012*). SKAP plus-end tracking is required for proper interactions of astral microtubules with the cell cortex and metaphase spindle positioning, but not for chromosome segregation (*Kern et al., 2016*). Second, SKAP binds directly to microtubules, and mutants that precisely disrupt microtubule binding result in dramatic defects in chromosome alignment and segregation (*Friese et al., 2016*; *Kern et al., 2016*).

In an important contrast to other components of the kinetochore-microtubule interface, the Astrin-SKAP complex associates dynamically with kinetochores during mitosis and only localizes to aligned and bi-oriented kinetochores (*Friese et al., 2016*; *Mack and Compton, 2001*; *Manning et al., 2010*; *Schmidt et al., 2010*). Thus, the Astrin-SKAP complex is the final component of the outer kinetochore that is recruited prior to anaphase onset. This unique localization timing suggests a model in which the Astrin-SKAP complex stabilizes kinetochore-microtubule attachments as cells prepare for chromosome segregation. However, the specific function and interactions for the Astrin-SKAP complex at kinetochores remain largely unknown.

Here, we identify a fourth subunit of the Astrin-SKAP complex, MYCBP, reconstitute the complete, four-subunit, human Astrin-SKAP complex from insect cells, and dissect its biochemical and cell biological interactions. Our work indicates that Astrin has a separable C-terminal kinetochore localization domain and an N-terminal region that associates with microtubules through its interaction with SKAP. Using cross-linking mass spectrometry to trap Astrin in its kinetochore-bound state, we find that the Astrin-SKAP complex interacts with the Ndc80 complex. We reconstitute an interaction between the N-terminal half of the Astrin-SKAP complex and the Ndc80 complex in the presence of microtubules. Importantly, we find that the Ndc80 and Astrin-SKAP complexes can bind to microtubules simultaneously to form an integrated interface. Our work suggests a model in which the Astrin-SKAP complex stabilizes correctly formed kinetochore-microtubule attachments through its own intrinsic microtubule binding activity and its coordinate association with microtubule-bound Ndc80 complex.

## Results

### Reconstitution of a 4-subunit Astrin-SKAP complex

To understand the molecular role of the Astrin-SKAP complex at kinetochores, we first sought to define the organization and assembly of the complex. As a first step to reconstituting an intact Astrin-SKAP complex, we began by reanalyzing the composition of the Astrin-SKAP complex isolated from human tissue culture cells. In addition to the previously defined subunits Astrin, SKAP, and LC8, we consistently identified MYCBP as an interacting partner (*Figure 1A*). In reciprocal immunoprecipitations, MYCBP isolated the Astrin-SKAP complex, as well as some of its previously defined interactors, including the AKAP family proteins and ARFGEF1 (*Furusawa et al., 2001*; *Furusawa et al., 2002*; *Ishizaki et al., 2006*; *Taira et al., 1998*) (*Figure 1A*). The mass spectrometry data obtained from these affinity purifications is consistent with MYCBP participating in multiple, distinct protein complexes, similar to our previous results with LC8 (*Schmidt et al., 2010*). Supporting its interaction as a component of the Astrin-SKAP complex, MYCBP localizes to kinetochores only after chromosome alignment (*Figure 1B*) and in an Astrin and SKAP-dependent manner (*Figure 1—figure supplement 1A*).

We next sought to reconstitute the complete 4-subunit Astrin-SKAP complex by co-expression in insect cells. We were able to isolate a complex containing full length versions of each protein that co-purified over Ni-NTA resin and size exclusion chromatography (*Figure 1C* and *Figure 1—figure supplement 1B*). To dissect the associations within the complex, we generated a series of truncations within Astrin. Truncating the C-terminal half of Astrin by removing residues 694–1193 (Astrin 1–693) did not alter the associations within the Astrin-SKAP complex and resulted in increased

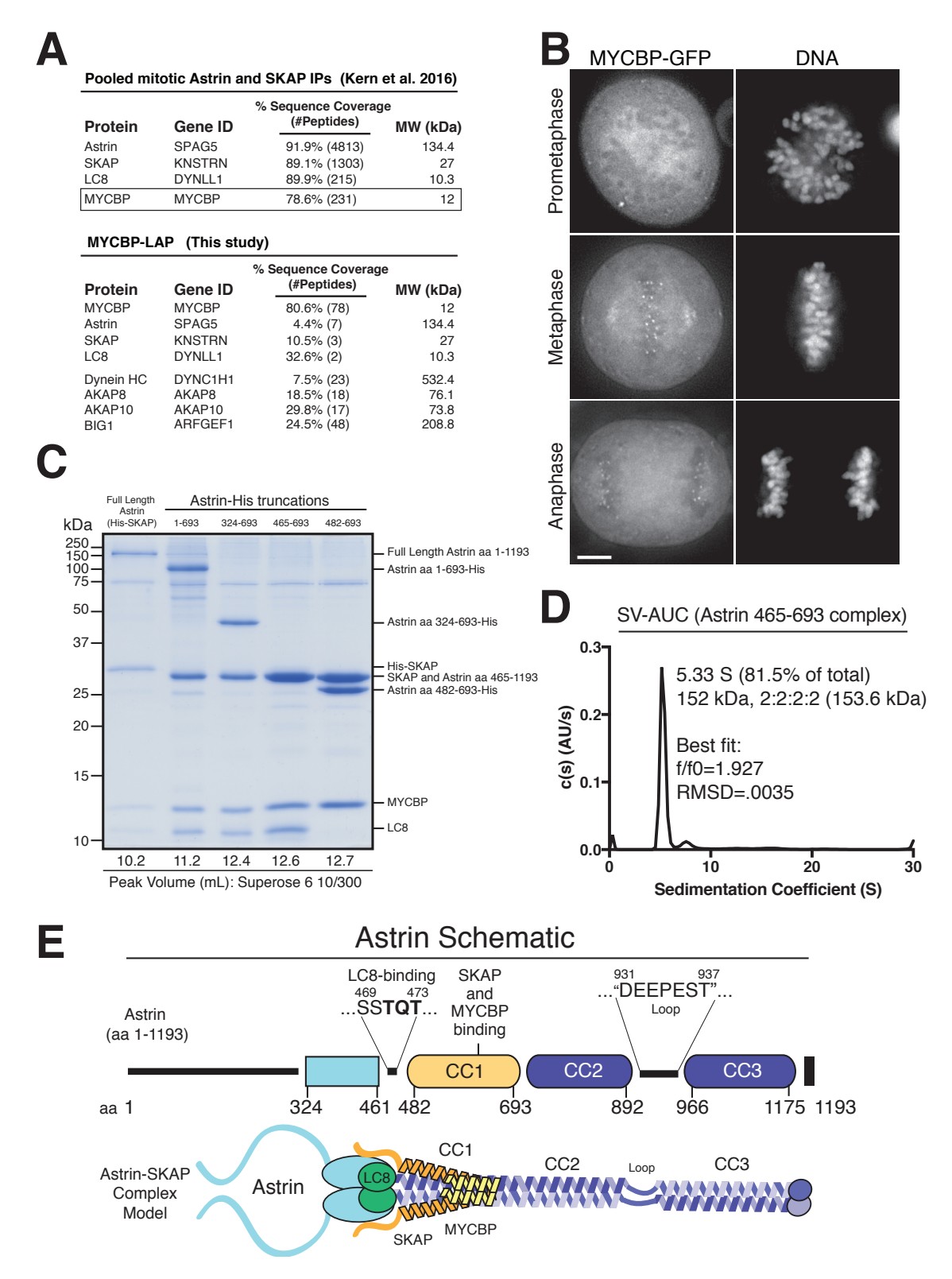

**Figure 1.** Reconstitution of a 4-subunit Astrin-SKAP complex. (**A**) Top: Pooled mass spectrometry data from Astrin and SKAP IPs (*Kern et al., 2016*) identifies MYCBP. Bottom: MYCBP IP (this study) isolates Astrin-SKAP complex components. (**B**) MYCBP-GFP localizes to the mitotic spindle and aligned kinetochores. Deconvolved image sections of selected MYCBP-GFP cells in mitosis were selected and scaled individually to show localization. Many cells from multiple experiments were analyzed with live and fixed cell microscopy to draw conclusions about MYCBP localization. Also see

*Figure 1 continued on next page*

*Figure 1 continued*

***Figure 1—figure supplement 1A**. Scale bar, 5 μm. (**C**) Coomassie gel of Astrin-SKAP complex purifications. Complex components and truncations were co-expressed using the MultiBac system in SF9 cells. Indicated His-tags were used for complex purification followed by gel filtration. For each complex, gel filtration peaks were pooled and spin concentrated before polyacrylamide gel loading. Gel filtration peak migration volumes are given below each lane (also see **Figure 1—figure supplement 1B**). Void Volume: 8.5 mL, Thyroglobulin (8.5 nm stokes radius) Size Standard: 13.1 mL. (**D**) Sedimentation velocity ultracentrifugation of the Astrin 465–693 complex. The complex was fit to a single major peak with the indicated statistics. (**E**) Schematic of Astrin-SKAP complex structure based on data in this figure and **Figure 1—figure supplement 1**.*

DOI: https://doi.org/10.7554/eLife.26866.002

The following figure supplement is available for figure 1:

**Figure supplement 1.** MYCBP kinetochore localization is dependent on Astrin and SKAP and Astrin-SKAP complex biochemical characterization.

DOI: https://doi.org/10.7554/eLife.26866.003

expression of a well-behaved complex (***Figure 1C*** and ***Figure 1—figure supplement 1B***). An additional truncation to remove the predicted N-terminal unstructured region of Astrin (residues 1–324; predictions made using Jpred: [***Drozdetskiy et al., 2015***]) further improved expression (Astrin 324–693), and maintained the association of all complex subunits (***Figure 1C***). In contrast, truncations within residues 324–465 resulted in aberrant (larger) gel filtration migration, consistent with altered structure or self-association (***Figure 1—figure supplement 1C***). Finally, we found that a region containing the central coiled-coil region of Astrin (residues 465–693) migrated appropriately by gel filtration and was sufficient to interact with SKAP, LC8, and MYCBP (***Figure 1C*** and ***Figure 1—figure supplement 1B***). Within this 465–693 region of Astrin, eliminating a small predicted loop (residues 465–482) containing a 'TQT' motif implicated in LC8 binding (***Lo et al., 2001***) disrupted the interaction between Astrin and LC8, but did not alter SKAP or MYCBP binding (***Figure 1C***). These truncated Astrin-SKAP complexes (Astrin 324–693 and Astrin 465–693) appeared as regular, elongated molecules by electron microscopy (***Figure 1—figure supplement 1D***).

To define the stoichiometry of the Astrin-SKAP complex, we conducted analytical ultracentrifugation (AUC) on the minimal Astrin 465–693 complex, as well as a version of this complex containing a GFP-tagged SKAP subunit. Fitting of the AUC data indicated a single major species was present (***Figure 1D*** and ***Figure 1—figure supplement 1E***). The molecular weight of this species fit to a complex with 2:2:2:2 stoichiometry (see Materials and methods).

Together, this work reveals the organization and stoichiometry of the Astrin-SKAP complex (***Figure 1E***) and provides the basis for a directed biochemical analysis of Astrin-SKAP interactions.

## The Astrin-SKAP complex binds directly to microtubules

We and others demonstrated previously that the microtubule binding activity of the Astrin-SKAP complex plays an important role in chromosome segregation (***Friese et al., 2016***; ***Kern et al., 2016***). To analyze the microtubule binding activity of the intact Astrin-SKAP complex, we conducted microtubule co-sedimentation assays (***Figure 2A***). The complex containing the N-terminal Astrin region (1-693) bound to microtubules with an apparent affinity of 3.2 μM (***Figure 2A,B***). Previous work from our lab and others indicated that SKAP contains the primary microtubule binding activity for the Astrin-SKAP complex (***Friese et al., 2016***; ***Kern et al., 2016***). Consistent with these results, the Astrin 324–693 and 465–693 complex constructs, which contain SKAP, also displayed microtubule binding in vitro (***Figure 2—figure supplement 1***). Furthermore, the Astrin-SKAP complex decorated and bundled microtubules based on negative stain transmission electron microscopy (***Figure 2C***), although it did not display an apparent ordered binding behavior. In contrast, we found that a charge swap mutant in SKAP ('5xD'), which substantially reduced Astrin-SKAP complex spindle microtubule localization in cells (***Kern et al., 2016***), also eliminates the binding of the Astrin-SKAP complex to microtubules in vitro (***Figure 2A***). These data demonstrate that the reconstituted Astrin-SKAP complex binds directly to microtubules.

## The Astrin C-terminal region targets the Astrin-SKAP complex to kinetochores

Astrin displays dynamic localization to kinetochores, localizing to bi-oriented kinetochores during metaphase and persisting into anaphase (***Figure 3A***; ***Figure 3—video 1***). Our biochemical analysis above defined regions of Astrin responsible for its different interactions. We next analyzed the

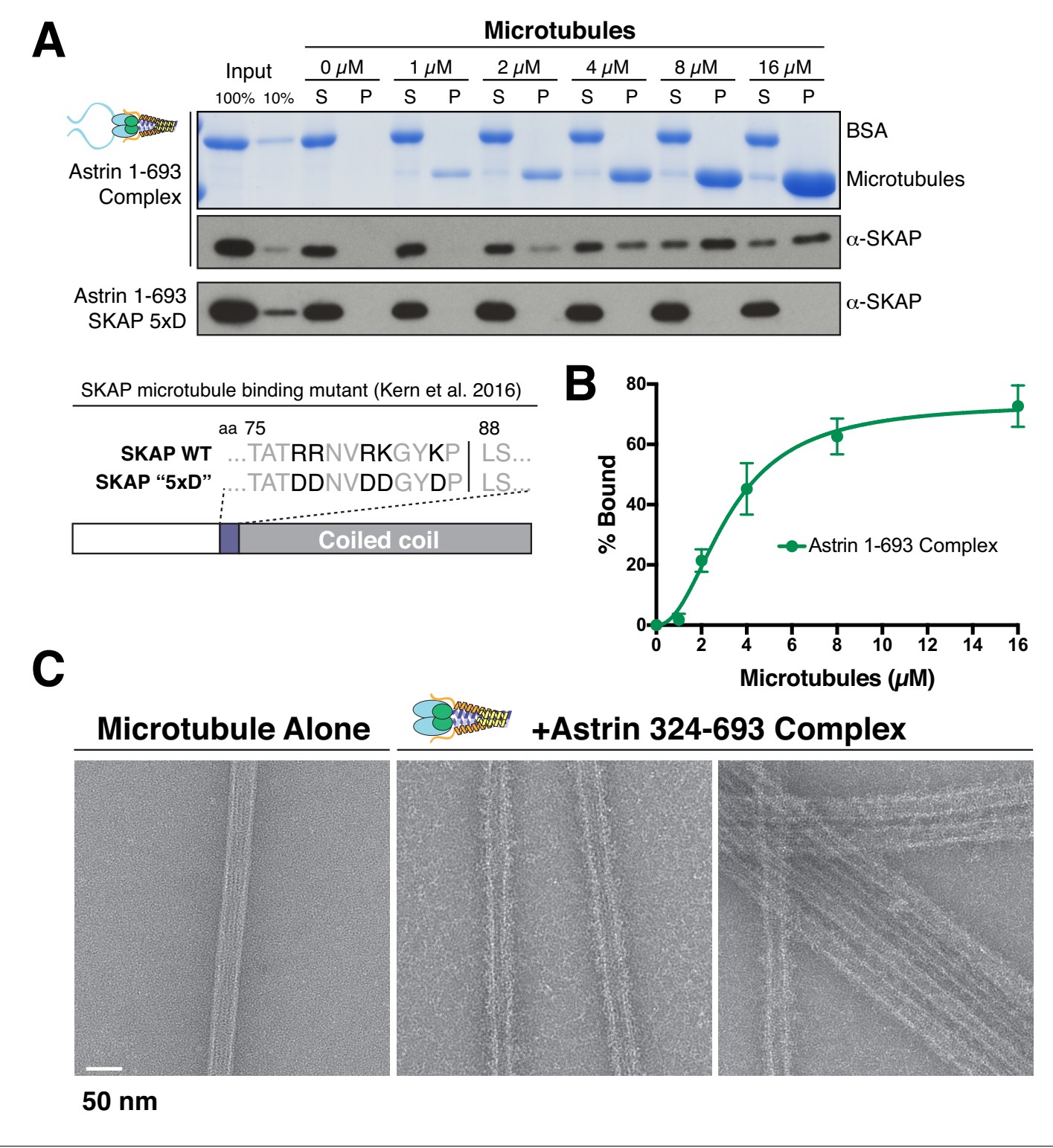

**Figure 2.** The Astrin-SKAP complex binds to microtubules through its SKAP microtubule-binding domain. (**A**) Top: Astrin 1–693 complex microtubule binding assays showing the Coomassie stained gel and α-SKAP Western blots from an example experiment. Bottom: Equivalent assay for the SKAP 5xD mutant version of the complex with the α-SKAP Western blot shown. Mutated residues for the 5xD mutant are illustrated below (also see [*Kern et al., 2016*]). (**B**) Microtubule-binding curve generated from triplicate binding experiments. Mean and standard deviation are plotted from quantified Western blots (see Materials and methods). The data was fit in Prism using the model of specific binding with a Hill slope. (**C**) Images of Astrin-SKAP complex-bound microtubules visualized with negative-stain electron microscopy.

*Figure 2 continued on next page*

*Figure 2 continued*

DOI: https://doi.org/10.7554/eLife.26866.004

The following figure supplement is available for figure 2:

**Figure supplement 1.** Astrin truncations that associate with SKAP are sufficient to bind microtubules in vitro.

DOI: https://doi.org/10.7554/eLife.26866.005

behavior of these truncations for Astrin localization in human tissue culture cells. In cells, the Astrin N-terminus (aa 1–693) localized to microtubules (*Figure 3B*), consistent with its ability to interact with the SKAP microtubule binding subunit (*Figure 1C*). However, Astrin 1–693 localized only weakly to kinetochores. In contrast, the Astrin C-terminus (aa 694–1193), which lacks the SKAP binding site, localized to kinetochores, but not microtubules (*Figure 3B*). The kinetochore localization of the Astrin C-terminus was only observed following the depletion of endogenous Astrin (*Figure 3B*), potentially due to competition between full length Astrin and the C-terminal construct for limited kinetochore binding sites. Notably, similar to full length Astrin, the Astrin C-terminus only localized to bi-oriented kinetochores (*Figure 3—figure supplement 1*). As predicted based on the inability of the Astrin C-terminus to bind SKAP, cells in which the Astrin C-terminus replaced endogenous Astrin displayed dramatically reduced localization of SKAP to kinetochores (*Figure 3C*).

To test the contributions of these regions to chromosome segregation, we developed a replacement strategy using a CRISPR/Cas9-based inducible knockout targeting Astrin (see [*McKinley and Cheeseman, 2017*]). Depletion of Astrin using the inducible knockout system resulted in a pronounced increase in mitotic cells with misaligned chromosomes and multipolar spindles (*Figure 3D*), hallmarks of defective kinetochore function. These defects were rescued by expression of full length Astrin, but not by expression of the Astrin 694–1193 truncation (*Figure 3D*). However, a shorter Astrin truncation (465–1193), which preserves the SKAP interaction, rescued these major chromosome segregation defects (*Figure 3D*). Thus, Astrin acts as the primary kinetochore targeting subunit of the Astrin-SKAP complex and the Astrin C-terminus contains a critical kinetochore localization domain. However, proper chromosome segregation additionally requires formation of the intact Astrin-SKAP complex and the kinetochore recruitment of the microtubule binding SKAP subunit.

## The Astrin-SKAP complex interacts with the Ndc80 complex in cells

Despite a well-defined role in kinetochore function, the kinetochore interaction partners for the Astrin-SKAP complex remain unclear. To identify kinetochore associations for this dynamically localized complex, we developed a protocol to trap transient interaction states by treating cells with the chemical crosslinking agent formaldehyde prior to affinity purification (*Figure 4A*; see Materials and methods). We found that this strategy was effective in isolating a chromatin-bound, and therefore kinetochore-localized population of protein based on the centrifugation of the cross-linked DNA during the purification (*Figure 4—figure supplement 1A*). Using this method, we were able to isolate known interactions between outer kinetochore components and their upstream kinetochore receptors (*Figure 4—figure supplement 1B*).

Although Astrin/SKAP preferentially localizes to bi-oriented kinetochores, Astrin will accumulate at kinetochores during an extended arrest in the Eg5-inhibitor STLC to form monopolar spindles (*Schmidt et al., 2010*). We took advantage of this behavior to generate a homogenous population of mitotic cells with microtubule-bound kinetochores and localized Astrin (*Figure 4B*). We then utilized an anti-Astrin antibody for immunoprecipitation. In the absence of crosslinking agent, our affinity purifications isolated the Astrin-SKAP complex, but not other kinetochore components (*Figure 4C*) consistent with previous studies (*Dunsch et al., 2011*; *Kern et al., 2016*; *Schmidt et al., 2010*). In contrast, when we used our cross-linking protocol, we additionally isolated the kinetochore-localized Ndc80 complex (*Figure 4C*), the core component of the kinetochore-microtubule interface (*Cheeseman et al., 2006*; *DeLuca et al., 2006*). Peptides for the Ndc80/Hec1 and Nuf2 subunits in particular were identified across multiple replicates of this experiment (*Figure 4—figure supplement 1C*). Thus, the Astrin-SKAP complex is closely associated with the Ndc80 complex where is it well positioned to act at the kinetochore-microtubule interface.

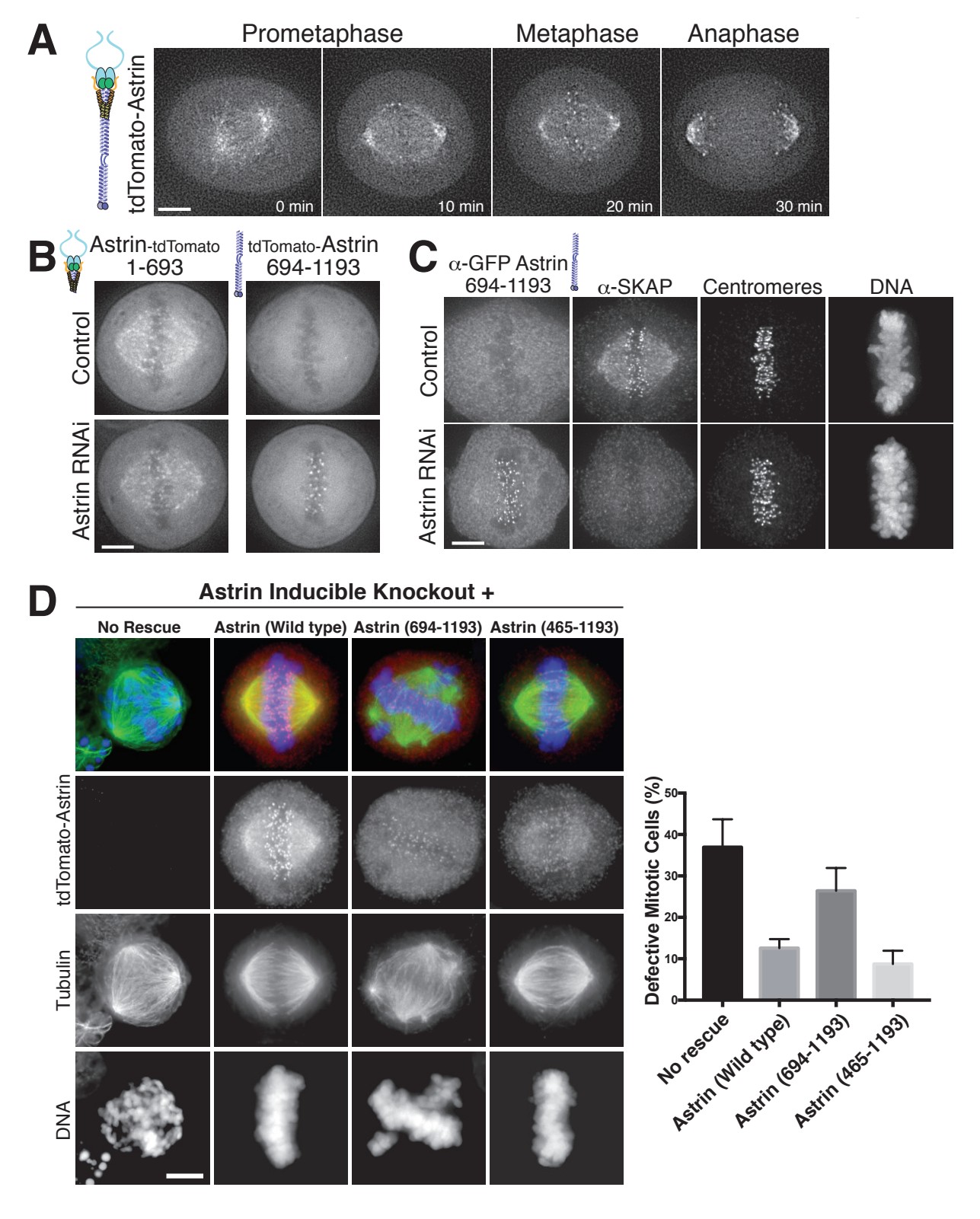

**Figure 3.** The Astrin-SKAP complex docks at kinetochores using two distinct domains. (A) Time lapse images showing localization of tdTomato-Astrin, introduced with the BacMam system into HeLa cells, during a mitotic time course (see *Figure 3—video 1*). Images represent deconvolved sections. (B) Localization of the indicated Astrin constructs, introduced with the BacMam system, in control or Astrin-depleted cells. Images are deconvolved sections and are scaled equivalently for each construct. (C) Immunofluorescence images for control or Astrin-depleted cells expressing GFP-Astrin 694–

*Figure 3 continued on next page*

eLIFE Research article

Biochemistry | Cell Biology

1193 with the indicated antibodies. Each image is a deconvolved and projected image stack through the mitotic spindle. Cells were scaled equivalently for each antibody, and DNA (Hoechst-stain) was scaled individually. Fixation (PF). (**D**) Astrin inducible knockouts with BacMam replacements. Left: deconvolved and projected immunofluorescence images of representative phenotypes for the indicated conditions. Images were scaled equivalently for Astrin antibody and individually for DNA (Hoechst-stain) and microtubules. Right: Quantification of defective mitotic cells from two experiments per condition (200 cells counted per experiment) showing the percent of cells with mis-aligned chromosomes and/or multi-polar spindles. Scale bars, 5 μm.

DOI: https://doi.org/10.7554/eLife.26866.006

The following video and figure supplement are available for figure 3:

**Figure supplement 1.** Astrin C-terminal domain (amino acids 694–1193) localizes to aligned kinetochores.

DOI: https://doi.org/10.7554/eLife.26866.007

**Figure 3—video 1.** Astrin localizes to aligned kinetochores during mitosis.

DOI: https://doi.org/10.7554/eLife.26866.008

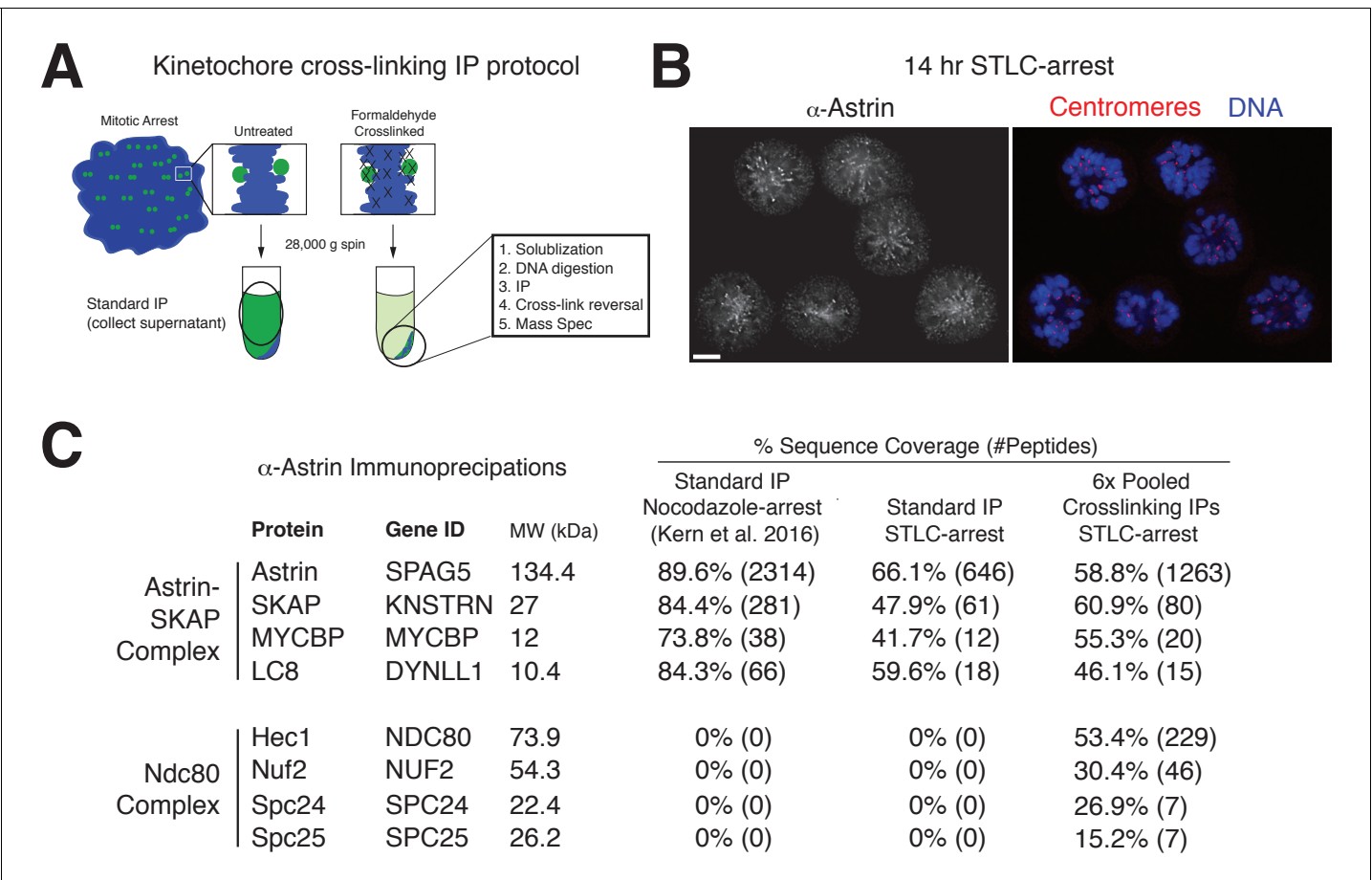

**Figure 4.** Kinetochore cross-linking mass spectrometry identifies Ndc80 as an Astrin-SKAP complex interaction partner. (**A**) Schematic of kinetochore cross-linking IP protocol (see Materials and methods). (**B**) Immunofluorescence of STLC-arrested HeLa cells showing Astrin localization to kinetochores using the α-Astrin antibody utilized for IPs. Fixation (M). Scale bar, 5 μm. (**C**) α-Astrin immunoprecipitation-mass spectrometry with the indicated arrests. The cross-linking IPs (right), shown as combined data from six independent experiments, reveal an interaction between the Astrin-SKAP and Ndc80 complexes. See *Figure 4—figure supplement 1* for additional data from the individual preparations and the Source dataset-Mass Spectrometry Data Supplement for the complete search information.

DOI: https://doi.org/10.7554/eLife.26866.009

The following figure supplement is available for figure 4:

**Figure supplement 1.** Kinetochore cross-linking mass spectrometry reveals kinetochore binding sites for dynamic interactions.

DOI: https://doi.org/10.7554/eLife.26866.010

# The Astrin-SKAP complex displays synergistic microtubule binding with the Ndc80 complex

Based on the interaction between the Astrin-SKAP and Ndc80 complexes observed in the crosslinking mass spectrometry, we next analyzed their interactions in vitro. We were unable to detect evidence for interactions between the soluble Astrin-SKAP and Ndc80 complexes (data not shown). As both the Ndc80 and Astrin-SKAP complexes bind to microtubules, and the Astrin-SKAP complex shows its strongest localization at attached kinetochores, we tested whether they altered each other's microtubule binding capabilities. Recent work found that a SKAP peptide (SKAP aa 57–147), containing the microtubule-binding domain, displayed competitive microtubule binding with the Ndc80 complex (*Friese et al., 2016*). In contrast, we found that the Astrin-SKAP complex displayed enhanced binding to microtubules in the presence of 1 µM Ndc80 Bonsai complex (*Figure 5A*), an internally truncated and shortened complex (*Ciferri et al., 2008*). We observed an apparent $K_D$ of ~3.2 µM for the Astrin-SKAP complex alone vs. 1.4 µM in the presence of the Ndc80 Bonsai complex. This altered binding requires a direct interaction between SKAP and the microtubule, as the SKAP 5xD mutant complex failed to bind to microtubules even in the presence of the Ndc80 complex (*Figure 5—figure supplement 1A*). We also observed enhanced apparent binding of the Astrin-SKAP complex in the presence of the Ndc80 Broccoli complex (*Figure 5—figure supplement 1B*), a version of the Ndc80 complex that contains all of the microtubule proximal regions, but is truncated to lack the 'roots' of the Ndc80 complex that target it to kinetochores (*Schmidt et al., 2012*). Additionally, we tested binding of the Astrin-SKAP complex to microtubules in the presence of an Ndc80 calponin homology domain (CH Domain) (aa 1–201) (*Hiruma et al., 2015*). Even though this construct binds microtubules more poorly than the complete Ndc80 complex, it was also able to increase the apparent microtubule binding of the Astrin-SKAP complex (*Figure 5—figure supplement 1C*), suggesting that the Ndc80 1–201 region contains the interaction site.

We next tested the ability of the Ndc80 and Astrin-SKAP complexes to associate simultaneously with microtubules. For these experiments, we varied the concentration of the Ndc80 Bonsai complex over a range from 0 to 6 µM in the presence of a constant 2 µM microtubules, allowing us to create conditions where the Ndc80 complex fully saturated the microtubule lattice (*Figure 5B*). Under these conditions, the Astrin-SKAP complex showed a concentration-dependent increase in binding in the presence of the Ndc80 complex (*Figure 5B*). Interestingly, even at the highest saturating concentrations of Ndc80 complex, the Astrin-SKAP complex was still able to bind robustly to the microtubule-Ndc80 complex, indicating that these complexes are able to bind to microtubules simultaneously. These results support synergistic, rather than competitive microtubule binding.

To assess the specificity and requirements for the interaction between the Astrin-SKAP and Ndc80 complexes in the context of microtubules, we next tested altered versions of each complex. We first tested the interaction between the human Astrin-SKAP complex and a 'Broccoli' version of the NDC-80 complex from *C. elegans*, an organism that lacks a detectable Astrin-SKAP complex homologue. Like the human complex, the *C. elegans* NDC-80 complex serves as a core microtubule interactor, but these proteins show distinct binding modes and significant sequence differences (*Wilson-Kubalek et al., 2016*). At low NDC-80 concentrations, we observed increased Astrin-SKAP complex microtubule binding (*Figure 5B*). As the NDC-80 complex promotes microtubule bundling (*Cheeseman et al., 2006*), we speculate that it may create higher avidity sites for the multimeric Astrin-SKAP complex. However, in contrast to the cooperative interactions with the human Ndc80 complex, we found that the *C. elegans* NDC-80 complex displayed competitive interactions with the Astrin-SKAP complex at high NDC-80 concentrations (*Figure 5B*). Therefore, the synergistic interaction we observed requires sequences or features specific to the human Ndc80 complex.

We also found that the N-terminus of Astrin is required to achieve robust interactions with the Ndc80 complex. Removing the Astrin N-terminal domain (1-464) significantly compromised its interaction with the Ndc80 complex as demonstrated by two observations. First, the Astrin 465–693 complex displayed only a modest enhancement of its microtubule binding activity in the presence of 1 µM Ndc80 complex ($K_D$ ~2.2 µM vs. 1.7 µM; *Figure 5C*). Second, the Astrin 465–693 complex was competed off of microtubules at increasing Ndc80 complex concentrations (*Figure 5D*). Thus, we have identified specific features on both the Ndc80 and Astrin-SKAP complexes required for their robust interaction in the context of microtubules and for their ability to bind simultaneously to create an integrated interface.

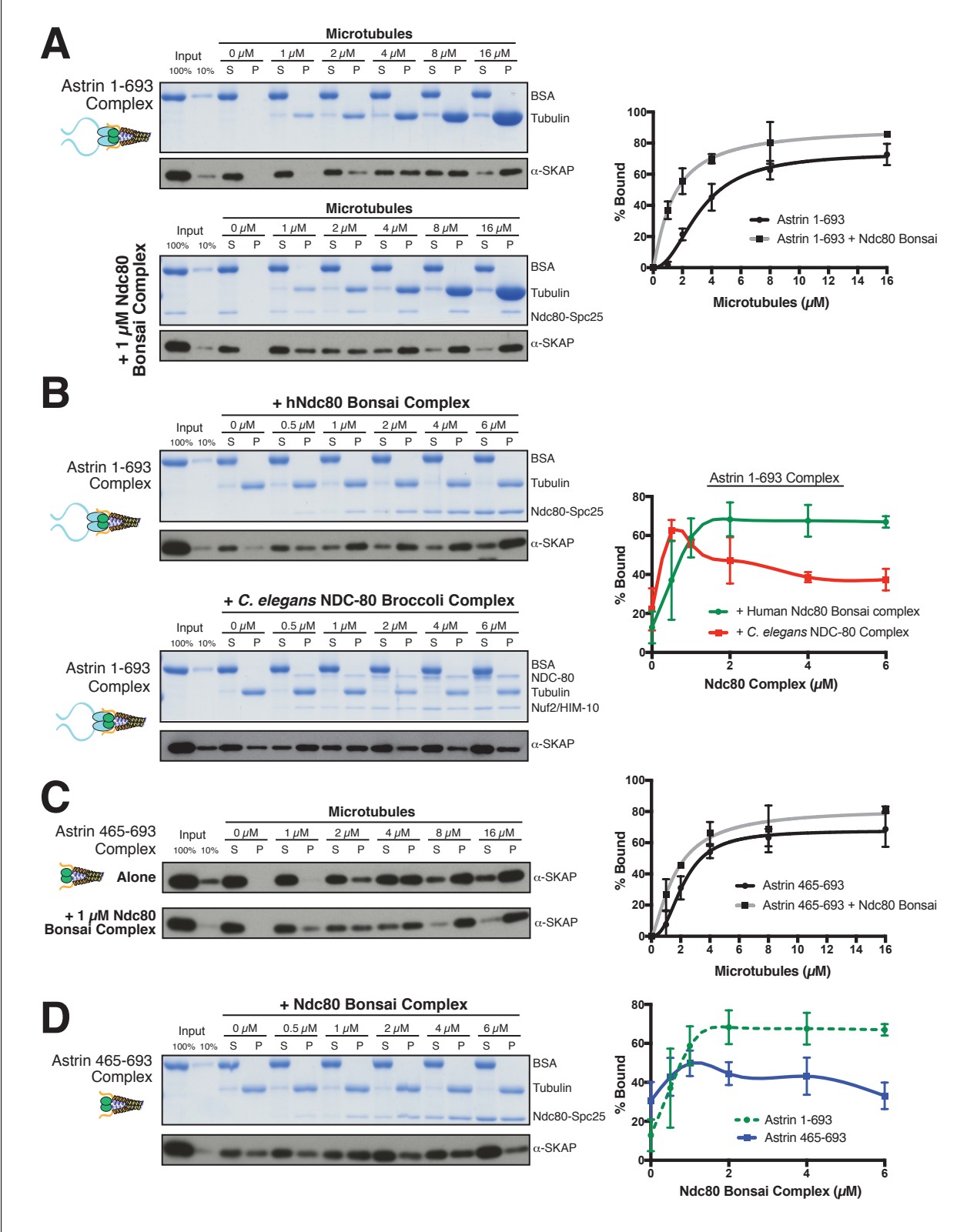

**Figure 5.** The Astrin-SKAP complex interacts with the Ndc80 complex in the presence of microtubules. (**A**) Left: Example of Coomassie-stained gel and western blot (the same gel was cut to generate each for loading comparison) for the Astrin 1–693 complex bound to microtubules in the absence (top) and presence of 1 μM Ndc80 Bonsai complex. Right: Quantification of Astrin-SKAP microtubule binding from triplicate experiments. Curve for microtubule binding without Ndc80 was reproduced from *Figure 2B*. Mean and standard deviation are plotted, and the data was fit with Prism using

*Figure 5 continued on next page*

*Figure 5 continued*

the model of specific binding with a Hill slope (**B**) Left: Ndc80 complex saturation experiment example for the Astrin 1–693 complex with human Ndc80 Bonsai complex (top) and the *C. elegans* Ndc80 Broccoli complex (bottom). Microtubules, 2 µM. Right: Quantification of binding from triplicate (human) or duplicate (*C. elegans*) experiments. Mean and standard deviation are plotted, and the trend line was generated from a coarse LOWESS fit in prism. (**C**) Left: Western blots of microtubule binding (as in part A) with the Astrin 465–693 complex. Right: Quantification as in part A, Curve for microtubule binding without Ndc80 was reproduced from *Figure 2—figure supplement 1B*. (**D**) Left: Ndc80 Bonsai complex saturation experiment for the Astrin 465–693 complex. Microtubules, 2 µM. Right: Quantification of binding from triplicate experiments as in (**B**). The curve from part B for the Astrin 1–693 complex is duplicated as a dotted line for comparison.

DOI: https://doi.org/10.7554/eLife.26866.011

The following figure supplement is available for figure 5:

**Figure supplement 1.** Astrin-SKAP microtubule binding is necessary for its co-pelleting with the Ndc80-microtubule complex and the Astrin-SKAP complex interacts with the microtubule-binding Ndc80 CH-domain.

DOI: https://doi.org/10.7554/eLife.26866.012

Together, these data suggest that the Astrin-SKAP and Ndc80 complexes co-assemble to form a stable interaction with microtubules. Upon the formation of bi-oriented kinetochore-microtubule interactions, the Astrin-SKAP complex is targeted to kinetochores through the Astrin C-terminus. This concentrates the Astrin-SKAP complex in the vicinity of the Ndc80 complex where it utilizes both the SKAP microtubule binding activity and the Astrin N-terminus to generate a coordinated interaction with the microtubule-binding interface of the Ndc80 complex. Both of these interactions likely contribute to stabilizing bi-oriented kinetochore-microtubule interactions, although our analysis of the Astrin replacement mutants (*Figure 3D*) suggests that the intrinsic SKAP microtubule binding activity plays the most critical role in this process.

## Discussion

The goal of mitosis is to ensure that each pair of sister chromatids forms bi-oriented attachments to the mitotic spindle. However, it remains unclear how kinetochore-microtubule attachments are modulated to ensure that only correct, bi-oriented attachments are stabilized, whereas incorrect attachments are eliminated. Prior work has focused on the negative regulation of improper kinetochore-microtubule attachments by Aurora B (*Lampson and Cheeseman, 2011*) and the role of force in stabilizing microtubule attachments (*Akiyoshi et al., 2010*). Our work suggests an additional potential mechanism to stabilize proper kinetochore-microtubule attachments through the activity of the Astrin-SKAP complex binding to microtubules and stabilizing the Ndc80-microtubule interface. In contrast to other established components of the kinetochore-microtubule interface, the Astrin-SKAP complex displays a unique localization timing to bi-oriented kinetochores (*Figure 3A*, *Figure 3—video 1*, and *Figure 6A*). Its switch-like kinetochore localization is the mirror opposite of the spindle assembly checkpoint components that target preferentially to unattached and mis-aligned kinetochores. This late mitotic localization brings the Astrin-SKAP complex to correctly attached kinetochores at a time of high kinetochore tension, just prior to chromosome separation and segregation at anaphase onset. Thus, the Astrin-SKAP complex may act to stabilize proper kinetochore-microtubule attachments to ensure correct chromosome segregation during metaphase and anaphase.

The Astrin-SKAP complex plays a critical role in chromosome segregation in human cells based on both RNAi-based depletions (*Dunsch et al., 2011*; *Kern et al., 2016*; *Manning et al., 2010*) and CRISPR/Cas9-based inducible knockouts ([*McKinley and Cheeseman, 2017*]; this study). In particular, we and others have recently shown that the SKAP microtubule-binding activity is required for proper chromosome segregation (*Friese et al., 2016*; *Kern et al., 2016*). Here, we reconstitute and validate Astrin-SKAP microtubule binding in vitro. Astrin's interaction with Ndc80 in the presence of microtubules places the Astrin-SKAP complex as an integral part of the outer kinetochore-microtubule interface along with the Ska1 complex (*Figure 6B*). The Ndc80 complex is the core microtubule binding unit of the kinetochore and is widely conserved across eukaryotes. Intriguingly, although the Astrin-SKAP complex is conserved throughout vertebrates, SKAP and Astrin knockouts in mice and rats have shown that this complex is dispensable for viability in rodents (*Grey et al., 2016*; *Xue et al., 2002*). Given the Astrin-SKAP complex's established role in human cells, it is possible

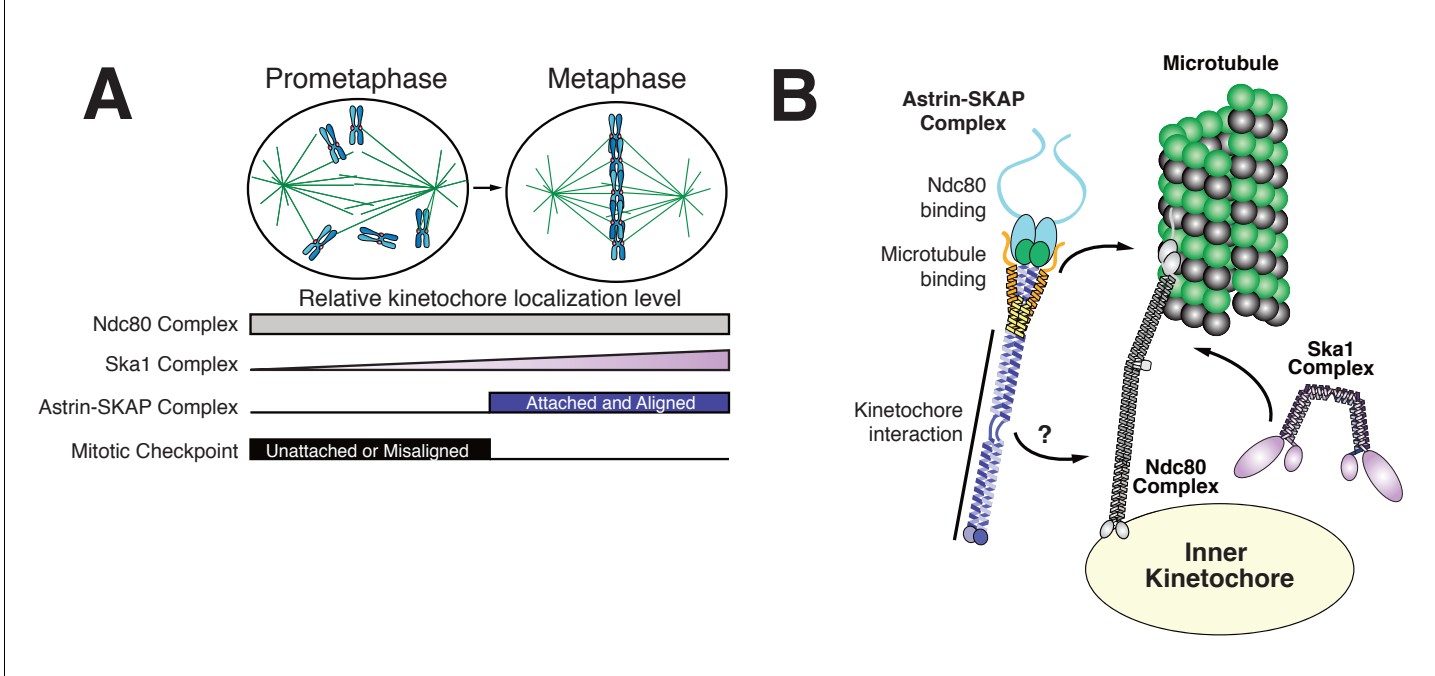

**Figure 6.** Model of Astrin-SKAP kinetochore attachment. (A) Schematic of kinetochore localization in prophase and metaphase for a selection of outer kinetochore components. Ndc80 complex localizes during mitosis to all kinetochores and the Ska1 complex kinetochore localization increases as mitosis proceeds. The Astrin-SKAP complex and the proteins of the mitotic checkpoint each show switch-like kinetochore localization, but with opposite timing. Astrin-SKAP complex localizes strongly to microtubule-bound and bioriented kinetochores, whereas mitotic checkpoint proteins localize preferentially to unattached and misaligned kinetochores. (B) The Astrin-SKAP complex binds to kinetochores through SKAP microtubule binding, Astrin's N-terminus binding to Ndc80 complex, and Astrin's C-terminus binding to an unknown kinetochore location. Through these multiple interactions, the Astrin-SKAP complex can provide an intra-kinetochore tether to bridge the kinetochore and stabilize microtubule interactions. Together, the Ska1, Ndc80, and Astrin-SKAP complexes form the major microtubule binding interface of metaphase human kinetochores.
DOI: https://doi.org/10.7554/eLife.26866.013

that the requirements for the Astrin-SKAP complex may be an important mitotic distinction between rodent and human kinetochores (*Akhmanova and van den Heuvel, 2016*).

Previous work analyzed truncations to narrow down Astrin's kinetochore binding domain to residues 482–1193 (*Dunsch et al., 2011*) and SKAP binding domain to residues 482–850 (*Friese et al., 2016*). Here, we further refine the SKAP binding site on Astrin using complex reconstitution and cell biological approaches to residues 482–693. We additionally identify a C-terminal Astrin domain (amino acids 694–1193) that is sufficient for kinetochore localization. Importantly, this region lacks the binding sites for the other components of the complex. Consequently, we find that this Astrin mutant is unable to recruit SKAP to kinetochores, and that cells containing only Astrin 694–1193 have chromosome segregation phenotypes similar to a Astrin knockout or SKAP depletion (*Kern et al., 2016*). Together with previous studies highlighting the importance of SKAP microtubule binding (*Friese et al., 2016*; *Kern et al., 2016*), these results establish the importance of kinetochore-localized Astrin-SKAP complex microtubule binding for proper chromosome segregation. The Astrin-SKAP complex has the potential to make several important contributions to microtubule interactions. First, the Astrin-SKAP complex is a multimeric microtubule binder due to the presence of two copies of the SKAP microtubule binding subunit per complex as defined by our analysis of Astrin-SKAP complex stoichiometry. This may serve to enhance the affinity of the kinetochore-microtubule interaction at a time when it is experiencing high force (*Dumont et al., 2012*; *Rago and Cheeseman, 2013*). Second, Astrin's interaction with the Ndc80 complex in the context of microtubules leads to the intriguing possibility that the Astrin-SKAP complex acts as a stabilizer or clamp at aligned kinetochores. The human Ndc80 complex has a well-defined binding site and angle on microtubules and is able to bind at every tubulin monomer within the microtubule lattice using the 'toe' region of its CH domain. In addition, the Ndc80 complex interacts with the negatively-charged

E-hook of tubulin using its positively charged N-terminal tail (*Alushin et al., 2012*, *2010*) (*Wilson-Kubalek et al., 2016*). Therefore, the presence of saturating Ndc80 complex on microtubules would be expected to thoroughly compete with and block most microtubule-associated proteins (MAPs). Indeed, *Friese et al. (2016)* recently noted that a small piece of SKAP, containing its microtubule binding domain, reciprocally competes with the Ndc80 complex for microtubule binding. In contrast to this expectation, we find that the reconstituted Astrin-SKAP complex binds even to Ndc80-saturated microtubules and that this activity requires SKAP microtubule binding. Similar to Ndc80's N-terminal tail, SKAP's microtubule binding domain is positively charged and predicted to be unstructured, so both likely require access to the tubulin E-hook. Astrin's interaction with Ndc80 may position SKAP to access the conformation of the Ndc80-bound E-hook (*Alushin et al., 2012*). Furthermore, in cells, SKAP may access microtubule regions beyond the reach of kinetochore-tethered Ndc80 (*Schmidt et al., 2010*). These results indicate that these two complexes can coexist and collaborate at kinetochores.

Due to the unique localization timing of the Astrin-SKAP complex, its regulation is an intriguing problem. Our work defines two Astrin-SKAP binding sites at kinetochores - the kinetochore-bound microtubule and the Ndc80 complex. These interactions can at least partly explain Astrin-SKAP's dependence on kinetochore-microtubule attachment for its localization. We propose a model in which Astrin-SKAP is fully active for kinetochore binding only when Ndc80 is docked on kinetochore microtubules. In this case, the Astrin-SKAP complex would act in the opposite manner of the Mps1 checkpoint protein, which has been shown to compete with microtubules for free Ndc80 complex (*Hiruma et al., 2015*; *Ji et al., 2015*). In addition to the microtubule-dependent interactions, we find that the C-terminus of Astrin contains a kinetochore interaction domain and localizes to kinetochores with similar localization timing to the full-length complex, despite an inability to bind to microtubules (*Figure 3*). This suggests additional regulation of the interaction between Astrin's C-terminus and its unknown kinetochore binding partner(s). We note that we sporadically obtained peptides from the CENP-L/N complex and the Dsn1 subunit of the Mis12 complex in our cross-linking immunoprecipitations of Astrin (*Figure 4—figure supplement 1C*). Reciprocal immunoprecipitations of the CENP-L-N complex also isolated the Astrin-SKAP complex (*Figure 4—figure supplement 1D*), supporting a potential association. Theoretically, Astrin's long coiled-coil domains would allow it to span large distances from these inner kinetochore locations to the microtubule interface. Alternatively, Astrin's C-terminus could bind to a secondary location on the Ndc80 complex. Defining the structure and kinetochore binding site of Astrin's C-terminal domain are important future goals for understanding Astrin-SKAP's kinetochore localization and regulation. Taken together, we have defined the 4-subunit Astrin-SKAP complex as an integrated component of the outer kinetochore microtubule interface with the Ndc80 complex.

## Materials and methods

### Cell lines and tissue culture

Cell lines were generated and maintained as described previously (*Kern et al., 2016*). For *Figure 1*, a MYCBP-LAP retroviral line was generated and HeLa cells were used for siRNA experiments. The HeLa cells used for these studies were originally obtained from Don Cleveland's lab (Ludwig Institute, La Jolla, CA). Experiments are conducted in the same parental cell line to facilitate comparisons. As the origin of the cells is not central to the nature of these experiments, we have not further validated their identity as HeLa cells (the most common contaminant anyway). HeLa Flp-In cell lines were used for experiments in *Figure 3C* using a parental line obtained from Stephen Taylor's lab (University of Manchester). For experiments using BacMams, HeLa cells were utilized. Cells are screened monthly to ensure that they are free of mycoplasma contamination.

### Protein depletion, induction, BacMam and Lentiviral expression

siRNA transfections were conducted as described previously (*Kern et al., 2016*) with ON-TARGETplus siRNAs (Dharmacon, Lafayette, CO)against MYCBP (Pool: 5'-AG GAGAAGCGUGCUGAAUA-3', 5'-GCCUAGAACUGGCCGAAAU-3', 5'-GUUGGUA GCCUUAUAUGAA-3', 5'-GAAGUAUGAAGCUA UUGUA-3'), Astrin (5'-CCAACUGA GAUAAAUGCU-3'; [*Manning et al., 2010*]), SKAP (5'-AGGC UACAAACCACUGAGUAA-3'; [*Dunsch et al., 2011*]), and LC8-1,2 (DYNLL1 Pool: 5'-GUUCAAAUC

UGGUUAAAAG-3', 5'-GAAGGACAUUGCGGCUCAU-3', 5'-GUACUAGUUUGUCGUGGUU-3', 5'-CAGCCUAAAUUCCAAAUAC-3'; DYNLL2 Pool: 5'-GGAAGGCAGUGA UCAAGAA-3', 5'-GACAA-GAAAUAUAACCCUA-3' 5'-CCAUGGAGAAG UACAAUAU-3', 5'-CAAAGCACUUCAUCUAUUU-3'; [*Schmidt et al., 2010*]). All analyses were conducted 48 hr after siRNA transfection. Flp-In cell lines were induced using 1 μg/ml doxycycline replaced every 24 hr (or as necessary). A modified baculovirus system to deliver constructs to mammalian cells (MultiBacMam; Geneva Biotech, Geneva, Switzerland) was used to introduce all tdTomato constructs. BacMam constructs were generated and used as described previously (*Kern et al., 2016*), except for those used in *Figure 3D* (see *Inducible CRISPR-Cas9 and Astrin mutant replacement* below). The BacMam virus was placed on cells for 6 hr and imaging was conducted ~24 hr post-addition. For *Figure 3—figure supplement 1*, cells expressing dtTomato-Astrin 694–1193 (cKC319) were generated using pKC242 (pLenti-based Vector).

## Antibody generation

Rabbit polyclonal antibodies for MYCBP (raised using His-MYCBP) and SKAP (raised using GST-SKAP aa 1–238 of the 'mitotic' - non-testes specific - SKAP isoform) were generated as described previously (*Kern et al., 2016*). The mitotic SKAP antibody was depleted for GST-specific antibodies using a GST column. Antibody affinity purification was performed as described previously (*Desai et al., 2003*), with the exception of a distinct coupling buffer (0.2 M Sodium Bicarbonate, 0.5 M NaCl, pH 8.3) used for the SKAP antibody.

## Immunofluorescence and microscopy

Immunofluorescence (IF) was described previously (*Kern et al., 2016*) unless noted in legend. IF was conducted with antibodies against SKAP (This study; 1:500), Astrin ([*Yang et al., 2006*]; mouse monoclonal antibody diluted from media for IF; *Figure 1—figure supplement 1*), Astrin ([*Kern et al., 2016*]; Rabbit polyclonal 1:2000), MYCBP (This study; 1:5000), LC8 (Abcam; 1:1000), Centromeres ('ACA' Antibodies Inc., Davis, CA; 1:100 or 1:300 for *Figure 3D* and *Figure 3—figure supplement 1B*), RFP (Rockland; 1:500) and GFP (Chromotek, Planegg-Martinsried, Germany; 1:200). The fixation (described in [*Kern et al., 2016*]) was conducted with PHEM/Formaldehyde (PF), pre-extraction with PHEM/Formaldehyde fixation (PEPF), or Methanol fixation (M), as indicated in the figure legends.

Microscopy was performed as described previously (*Kern et al., 2016*) using a DeltaVision Core microscope (GE Healthsciences, Pittsburg, PA). For the tdTomato-Astrin timecourse (*Figure 3A* and *Figure 3—video 1*), a single focal plane was chosen at the start of the movie and imaged at 2 min intervals. TRITC (tdTomato) and DAPI (Hoechst) exposure levels and time were chosen to be as limited as possible for adequate imaging to ensure mitotic progression and minimal photobleaching.

## Inducible CRISPR-Cas9 and astrin mutant replacement

An inducible Cas9 cell line targeting Astrin (cKC326) was generated as described in (*McKinley and Cheeseman, 2017*) using a cTT20 containing pKC251, guide RNA against Astrin: 5'-GACAAGGCAG TCAGATCTGG-3' (*Wang et al., 2014*). cKC326 cells were seeded on day '0' with 1 μg/mL doxycycline. On day one, media was changed for media with fresh 1 μg/mL doxycycline and BacMam constructs were added. Doxycycline was replenished the next day. On day three, cells were fixed in 4% paraformaldehyde in PBS at 37°C for 10 min, permeablized with PBS + 0.2% Triton X-100, and blocked in Abdil (20 mM Tris, 150 mM NaCl, 0.1% Triton X-100, 3% BSA, and 0.1% NaN₃, pH 7.5) for 0.5 hr. Rabbit anti-RFP Antibody Pre-adsorbed (Rockland, Limerick, PA) was added in Abdil (1:500) for 2 hr. 1 μg/ml Hoechst-33342 (Sigma, St. Louis, MO) was added with the secondary antibody mix. Cells were rinsed before incubation with Mouse anti-α-Tubulin−FITC antibody (Sigma, St. Louis, MO) for 1 hr followed by rinse and mounting to coverslip as described (*Kern et al., 2016*). Images were acquired on a Nikon eclipse microscope equipped with a CCD camera (Clara, Andor, Concord, MA) using a Plan Fluor 40x oil 1.3NA objective (Nikon, Melville, NY) and appropriate fluorescence filters and deconvolved using NIS-Elements AR (Nikon).

## Protein expression and purification

The human Ndc80 'Bonsai' complex was expressed and purified as described previously (*Ciferri et al., 2008*) with gel filtration post-GST cleavage into Column Buffer (20 mM HEPES, 150 mM KCl, 1 mM DTT, pH 7.5) on a Superose 6 10/300 GL column. GST-hNDC80 1–201 (CH-domain, [*Hiruma et al., 2015*]) was expressed and purified in the same manner. The *C. elegans* and human Ndc80 'Broccoli' complexes (*Schmidt et al., 2012*) were expressed in the same manner, purified using Ni-NTA resin, and gel filtered into Column Buffer. Peak fractions were pooled and spin concentrated using 10 kDa cutoff Vivaspin20 columns (GE Healthsciences, Pittsburg, PA) and 30 kDa cutoff Amicon Ultra-15 columns (EMD-Millipore, Billerica, MA) to ~4 mg/mL or needed concentrations. The Ndc80 Bonsai complex, CH Domain, and human Ndc80 Broccoli proteins were used fresh or frozen. *C. elegans* Ndc80 Broccoli was used fresh to maximize the accessible concentration. For freezing, aliquots were frozen in liquid nitrogen before storage at −80°C. Upon thawing, protein was centrifuged for 10 min at 21,000 x g and the concentration was re-tested prior to dilution for experiments.

For Astrin-SKAP complex expression, all four complex components (and utilizing the non-testis/mitotic version of SKAP; aa 1–238; see [*Kern et al., 2016*]) were iteratively cloned into the pACE-BAC1 (MultiBac; Geneva-Biotech, Geneva, Switzerland) vector with the indicated truncations and tags. Bacmids were generated from finished plasmids using DH10EmBacY cells according to manufacturer's instructions. *Spodoptera frugiperda* (Sf9) cells were cultured in Sf-900 III SFM (Life Technologies, Carlsbad, CA) and P1 virus was generated from cells transfected with Cellfectin II reagent (Life Technologies) according to manufacturer's instructions. P2 virus was then generated by infecting cells at 1.5 million cells/mL with P1 virus at an MOI ~0.1 with viral generation monitored using the YFP reporter and harvested at ~72 hr. P2 virus was then used to infect 500 mL to 1 L of SF9 cells at 2 million cells/mL at an MOI ~2–5. At 72 hr, infected cells containing expressed Astrin-SKAP protein were harvested by resuspending in PBS at half the cell volume and drop-freezing in liquid nitrogen.

For Astrin-SKAP complex purification, frozen cell pellets were resuspended (1:1 weight/volume) and thawed in insect lysis buffer (50 mM Sodium Phosphate, 300 mM NaCl, 60 mM Imidazole, pH 8) with 10 mM beta-mercaptoethanol, 1 mM PMSF, and one Complete Mini, EDTA-free tablet (Roche, Basel, Switzerland) added after thaw. Cells were then lysed by sonication and pelleted at 50,000 g for 35 min. Cleared supernatant (carefully avoiding any visible debris and re-pelleting as necessary) was then bound to 0.5 mL Ni-NTA resin for 2 hr at 4°C, followed by washing in Wash Buffer (50 mM Sodium Phosphate, 300 mM NaCl, 0.1% Tween-20, 40 mM Imidazole, pH 8). Protein was eluted with elution buffer (50 mM Sodium Phosphate, 300 mM NaCl, 250 mM Imidazole, pH 7) using a Poly-Prep Column (Bio-Rad, Hercules, CA). A 1 mL pool of the highest concentration elution fractions was then spun at 21,000 g for 10 min and supernatant loaded onto a Superose 6 10/300 GL Column into Column Buffer for gel filtration. Protein for electron microscopy with microtubules or the high concentration microtubule-binding assay was instead gel filtered into BRB80 buffer (80 mM PIPES pH 6.8, 1 mM MgCl2, and 1 mM EGTA) with 1 mM DTT added. Peak fractions were then pooled and spin concentrated using 10 kDa MWCO Vivaspin2 columns (GE Healthsciences, Pittsburg, PA). Fresh protein (never frozen) was used for all experiments with Astrin-SKAP complex. We note that full length Astrin complex, while competent for gel filtration, expresses at an especially low level, does not spin concentrate well, and precipitates in low salt conditions (BRB80 buffer with 50 mM KCl) and therefore was not used for microtubule binding experiments.

## Microtubule binding assays

Microtubule pelleting assays were conducted as described previously (*Cheeseman et al., 2006*) with taxol-stabilized microtubules. For Astrin-SKAP complex variants, concentrations were approximately matched using a Coomassie gel for a final concentration in the assay of ~5 nM (assuming 2:2:2:2 complex stoichiometry) and protein (stored on ice) was used within 5 days of initial purification. Replicates for experiments were performed using different protein mixes and often were conducted on different days, necessitating fresh preparations of microtubules and new thawed tubes of Ndc80 Bonsai complex. Ndc80 proteins were added to match the indicated final concentrations and 0.5 mg/mL BSA (final) was added to all assays. All protein mixes were designed such that the Column Buffer was diluted 6X in the final mix for a final salt concentration of 25 mM KCl in BRB80 buffer. For

all assays, the reaction of binding proteins with microtubules was incubated at room temperature for 10 min prior to loading onto glycerol cushion and pelleting. For experiments with increasing microtubule concentrations, 100% and 10% input samples were taken from a single protein master mix used for all concentrations. For experiments with increasing Ndc80 concentrations, the inputs were taken from the sample without added Ndc80. Experiments were analyzed using Acquastain (Bulldog Bio, Portsmouth, NH) to verify appropriate protein loading and experimental set up. Western blotting was conducted using the long SKAP antibody (*Schmidt et al., 2010*) at 1 µg/mL. The microtubule binding assays were quantified in Adobe Photoshop from scanned versions of the Western blots based on the integrated intensity over background for each sample. *Figure 1—figure supplement 1B* was quantified using Image Studio Lite.

## Analytical Ultracentrifugation

Sedimentation-velocity experiments for Astrin 465–693 complex constructs were conducted using purified protein at ~2 µM (see *Figure 1—figure supplement 1E*) in modified Column Buffer (DTT replaced with 0.5 mM TCEP), and using a Beckman Optima XL-I analytical ultracentrifuge (Beckman Coulter, Indianapolis, IN) in absorbance mode (MIT Biophysical Instrumentation Facility, Cambridge, MA). Data were collected at 20°C at 30,000 rpm. The data were fit using SEDFIT with a continuous sedimentation coefficient distribution model, assuming a single frictional coefficient. Molecular weights were estimated using the best-fit frictional coefficients. We note that the Stokes radius predicted by this fit was inconsistent with that determined by gel filtration (slightly larger than the 8.5 nm standard). When using the AUC S-value and the approximate gel filtration Stokes radius, the calculated fit is closer to a complex with 4 SKAP subunits, but 2 each of the remaining subunits. We chose the 2:2:2:2 complex fit as gel filtration migration of elongated proteins is subject to some amount of error versus globular size standards. However, we cannot exclude the possibility that there are more than 2 SKAP subunits per complex.

## Electron microscopy

Microtubules were prepared by polymerizing 5 mg/ml porcine brain tubulin (Cytoskeleton, Denver, CO) in polymerization buffer (80 mM PIPES, pH 6.8, 1 mM EGTA, 4 mM MgCl2, 2 mM GTP, 9% dimethyl sulfoxide) for 30 min at 37°C. Paclitaxel was added at 250 µM before further incubation of 30 min at 37°C. The polymerized microtubules were then incubated at room temperature for several hours before use. Microtubules (2.5 uM) diluted in BRB80 were absorbed for 1 min to carbon-coated, glow-discharged grids. 4 µL of the Astrin 324–693 complex (~250 nM complex in BRB80) was then added to the microtubules and after a short incubation the grid was blotted and negatively stained with 1% Uranyl Acetate. Single particles of both Astrin 324–693 and Astrin 465–693 complexes (~25 nM in Column Buffer) were incubated shortly on a glow discharged carbon-coated grids and stained with 1% Uranyl Acetate and Uranyl Formate respectively. Images were acquired on a FEI Tecnai TF20 200kV FEG Transmission Electron Microscope (TEM) (FEI, Hillsboro, Oregon), using a 4K × 4K Teitz (Gauting, Germany) camera.

## Kinetochore Cross-linking immunoprecipitation and mass spectrometry

Standard immunoprecipitation mass spectrometry was conducted as described previously (*Kern et al., 2016*). For cross-linking mass spectrometry, HeLa cells at ~70% confluency in 60 × 15 cm dishes cells were arrested in Nocodazole at 330 nM or STLC at 10 µM for 14 hr. The subsequent steps were then performed at room temperature: A mitotic shake-off was performed to harvest cells, cells were centrifuged at 1000 x g (1–3 mL of cells), and resuspended in 20 mL of either PBS (Nocodazole preps) or PHEM Buffer (STLC preps) plus 1.2% Formaldehyde. Formaldehyde cross-linking conditions were optimized based on (*Klockenbusch and Kast, 2010*). Formaldehyde was obtained from heating 16% paraformaldehyde (Electron Microscopy Sciences, Hatfield, PA) at 80°C for 2 hr and then passing through a 0.2 µm filter. Cells were gently rocked in fixative for ~7 or 17 min before centrifugation for 3 min at 1500 x g for a total fixation time of ~10 min or 20 min for Astrin IPs #2, 3, 5 (*Figure 4—figure supplement 1C*). The fixed cell pellet was then resuspended in 0.125 M Glycine in PBS for quenching and rocked for 10 min before centrifugation at 1500 x g. The quenched cell pellet was resuspended in an equal volume of HeLa Lysis Buffer and frozen as described previously (*Cheeseman and Desai, 2005*).

For the preparation, the frozen cell pellet was resuspended, lysed, and centrifuged at ~28,000 x g as described previously (*Cheeseman and Desai, 2005*). The pellet was then resuspended in 4 mL of LAP-CHIP Buffer (10 mM Tris, 50 mM KCl, 0.1% Sodium Deoxycholate, 0.5% N-lauroyl sarcosine, 10% Glycerol, pH 8) and 1 mM DTT and 10 µg/mL LPC (Leupeptin, Pepstatin A, Chymostatin: EMD Millipore, Billerica, MA) were added. The sample was then sonicated to further shear DNA (see *Figure 4—figure supplement 1A*). After sonication, Triton-X 100 (to 1%), $MgCl_2$ (to 1 mM), and 4 µl Benzonase (Sigma, St. Louis, MO) were added and the suspension was gently rotated at 4°C for 1.5 hr to digest the DNA and free the associated protein. Sample was then centrifuged at 13,000 x g for 10 min, the cleared supernatant was collected, and KCl was added up to 300 mM. The supernatant was incubated with the appropriate antibody-coupled beads (Rabbit: α-GFP for LAP-Mis12, α-Astrin [*Kern et al., 2016*], or anti-CENP-L [*McKinley et al., 2015*]) for 1 hr at 4°C, washed, and eluted as described previously (*Cheeseman and Desai, 2005*). Formaldehyde cross-links were reversed by heat treatment twice (95°C for 5 min): once after resuspension of precipitated protein elution and again after the tryptic digest.

## Acknowledgements

We thank members of the Cheeseman laboratory, the Brohawn laboratory, Laurel Wright, Priya Budde, and Ron Milligan for support, input, and critical reading of the manuscript. We thank Ian Whitney for technical assistance. Debby Pheasant and the Biophysical Instrumentation Facility for the Study of Complex Macromolecular Systems (NSF-0070319) are gratefully acknowledged. This work was supported by a Scholar award to IMC from the Leukemia and Lymphoma Society and a grant from the NIH/National Institute of General Medical Sciences to IMC (GM088313) and EWK and Ronald A Milligan (GM052468).

## Additional information

### Funding

| Funder | Grant reference number | Author |
|---|---|---|
| National Institute of General Medical Sciences | GM088313 | Iain M Cheeseman |
| Leukemia and Lymphoma Society | Scholar Award | Iain M Cheeseman |
| National Institute of General Medical Sciences | GM052468 | Elizabeth M Wilson-Kubalek |

The funders had no role in study design, data collection and interpretation, or the decision to submit the work for publication.

### Author contributions

David M Kern, Conceptualization, Data curation, Formal analysis, Investigation, Methodology, Writing—original draft, Writing—review and editing; Julie K Monda, Kuan-Chung Su, Investigation, Writing—review and editing; Elizabeth M Wilson-Kubalek, Data curation, Funding acquisition, Investigation, Methodology, Writing—review and editing; Iain M Cheeseman, Conceptualization, Formal analysis, Supervision, Funding acquisition, Methodology, Writing—original draft, Writing—review and editing

### Author ORCIDs

David M Kern http://orcid.org/0000-0001-8529-9045
Iain M Cheeseman http://orcid.org/0000-0002-3829-5612

### Decision letter and Author response

Decision letter https://doi.org/10.7554/eLife.26866.016
Author response https://doi.org/10.7554/eLife.26866.017

## Additional files

### Supplementary files

• Source data 1. Source dataset-Mass spectrometry data. Complete mass spectrometry searches using methods described in (*Washburn et al., 2001*) for affinity purification/mass spectrometry data sets described in this paper (data from this study; [*Kern et al., 2016*] [*Gascoigne et al., 2011*]). Individual Astrin cross-linking immunoprecipitations are listed based on the order in *Figure 4—figure supplement 1*. These samples have not been pruned for common or antibody-specific contaminants.

DOI: https://doi.org/10.7554/eLife.26866.014

• Transparent reporting form

DOI: https://doi.org/10.7554/eLife.26866.015

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
