## [Decision Letter]

Thank you for submitting your article "Astrin-SKAP complex reconstitution reveals its kinetochore interaction with microtubule-bound Ndc80" for consideration by *eLife*. Your article has been reviewed by three peer reviewers, and the evaluation has been overseen by a Reviewing Editor and Anna Akhmanova as the Senior Editor. The reviewers have opted to remain anonymous.

The reviewers have discussed the reviews with one another and the Reviewing Editor has drafted this decision to help you prepare a revised submission.

Kern et al., reanalyze Astrin localisation, identify MycBP as a subunit of the Astrin-SKAP complex, confirm regions of Astrin responsible for its localization and report the minimal domain in Astrin required for a four member Astrin-SKAP-MycBP-LC8 complex. Biochemical reconstitution of a truncated Astrin complex indicates 2:2:2:2 stoichiometry, and microtubule-interaction studies using this truncated complex show its cooperative interaction with truncated Ndc80 complex in the presence of stabilized microtubules.

The manuscript's original contribution lies in reconstituting a truncated four-member Astrin-SKAP complex and showing its stoichiometry; the stoichiometry of the complex is consistent with previous reports on Astrin from Musacchio and Hatzfeld groups (Friese et al., 2016, Gruber et al., 2002). The most important claim is that the truncated Astrin-SKAP complex and truncated Ndc80 complex bind synergistically to microtubules but the evidence to support this must be strengthened.

Essential revisions:

1) The authors should confirm their in vitro microtubule binding experiments using full length Ndc80 complexes, which they have shown that they can purify. In particular there are a few striking differences in the way in vitro microtubule binding assays are performed in this study compared to a previous study from Andrea Musacchio. While the Musacchio lab used the full length Ndc80 complex and fixed the Ndc80 concentration and varied the SKAP concentration in the MT binding assays, the Cheeseman lab used the Ndc80 bonsai and performed MT binding assays with fixed concentration of Astrin/SKAP and varied Ndc80 concentration. To convincingly compare the results, the MT binding assays need to be performed with full length Ndc80 complex and full length Astrin/SKAP complex and also by fixing Ndc80 and increasing Astrin/SKAP. This should unambiguously confirm the cooperative binding.

2) The authors should assay more thoroughly the requirement for synergistic binding in vivo for chromosome attachment, behaviour or segregation. This is important because Friese et al., Nat Comm. published that Ndc80 does not bind synergistically to microtubules. The authors could compare the behaviour of full length and the 456 truncation mutant, for example.

---

## [Author Response]

[…] The manuscript's original contribution lies in reconstituting a truncated four-member Astrin-SKAP complex and showing its stoichiometry; the stoichiometry of the complex is consistent with previous reports on Astrin from Musacchio and Hatzfeld groups (Friese et al., 2016, Gruber et al., 2002). The most important claim is that the truncated Astrin-SKAP complex and truncated Ndc80 complex bind synergistically to microtubules but the evidence to support this must be strengthened.

We appreciate the supportive comments regarding our complex reconstitution and additional studies. In addition to the comments below, we would like to mention some key features of these biochemical studies and the nature of the Astrin-SKAP complex composition/stoichiometry. Although we didn’t emphasize this in the text, reconstituting the Astrin-SKAP complex was perhaps the most complex biochemical challenge ever conducted by our laboratory. Successfully isolating this recombinant complex required us to first define the physiological features of the Astrin-SKAP complex, including defining the correct SKAP isoform (see Kern et al. 2016) and identifying a 4^th^ complex subunit (MYCBP – this paper). Even with this information, we needed to test multiple different expression systems and carefully optimize the purification procedure and buffer conditions for this very finicky complex. In speaking with the Gruneberg lab, who are excellent biochemists and are also interested in the Astrin-SKAP complex, they also experienced significant challenges in trying to reconstitute this complex that prevented them from conducting similar studies. Obtaining the full length Astrin-SKAP complex and the slightly truncated variant that lacks the Astrin kinetochore localization domain (but retains other key features) allowed us for the first time to test the key properties and behaviors of this intact complex, including its stoichiometry, microtubule binding behavior, and synergy with the Ndc80 complex.

The biochemical analysis from the Musacchio and Hatzfeld labs (Friese et al., 2016, Gruber et al., 2002) provides an important background and context for our work. However, we would particularly like to highlight the importance of reconstituting a full length and 4-subunit Astrin-SKAP complex containing all of the physiological cellular binding partners. For the Friese et al. paper, the Musacchio group primarily analyzed the stoichiometry of a shorter SKAP peptide. This suggested that SKAP formed a trimer, but it is harder to make strong statements regarding its stoichiometry in the absence of its physiological Astrin interaction. The version of the minimal Astrin-SKAP complex that the Musacchio lab obtained (lacking LC8 and MYCBP) was not well behaved enough for AUC. The Gruber et al. paper was performed with Astrin (with a small truncation) that was expressed in bacteria, denatured with urea, and refolded from inclusion bodies. Although this is an excellent paper, refolding a large >1000 amino acid protein from bacteria could result in anomalous behaviors. This is particularly true for a protein that lacks its native and constitutive binding partners (SKAP, LC8, and MYCBP). From our reading of the paper, the stoichiometry information that they report was largely based on the two putative N-terminal globular (“lollipop”) regions that they observe by EM. As both we and the Musacchio group note, the N-terminus of Astrin is predicted to be largely disordered, and may just have re-folded incorrectly into an aggregated structure. Thus, we believe that our study provides the definitive information on the nature and stoichiometry of the physiological Astrin-SKAP complex.

Essential revisions:1) The authors should confirm their in vitro microtubule binding experiments using full length Ndc80 complexes, which they have shown that they can purify. In particular there are a few striking differences in the way in vitro microtubule binding assays are performed in this study compared to a previous study from Andrea Musacchio. While the Musacchio lab used the full length Ndc80 complex and fixed the Ndc80 concentration and varied the SKAP concentration in the MT binding assays, the Cheeseman lab used the Ndc80 bonsai and performed MT binding assays with fixed concentration of Astrin/SKAP and varied Ndc80 concentration. To convincingly compare the results, the MT binding assays need to be performed with full length Ndc80 complex and full length Astrin/SKAP complex and also by fixing Ndc80 and increasing Astrin/SKAP. This should unambiguously confirm the cooperative binding.

We appreciate these points. For the revised manuscript, we have conducted substantial additional biochemistry and microtubule binding assays to address these issues.

To consider these experiments, we would first like to highlight the substantial biochemical constraints for working with these complex and challenging proteins. The Musacchio lab (Friese et al. 2016) was able to obtain very high concentrations (80 µM) of a short SKAP peptide (~100 amino acids) to use for their competition experiment. This peptide lacks the vast majority of protein features present in the intact Astrin-SKAP complex. We do not doubt the biochemical validity of their competition experiments with this construct, in which two minimal microtubule binding regions from Ndc80 and SKAP would be expected to compete. However, our goal was to assess whether SKAP behavior accurately reflects the activity of the entire Astrin-SKAP complex, in which additional interactions would be feasible.

We have endeavored to use the full Astrin-SKAP dimeric complex for our in vitro work, to better match the situation in a cell. We were incredibly excited to be able to express and purify the full length Astrin-SKAP complex. However, this full length complex expresses at very low levels (<200 nM) and we have only been able to isolate small quantities. In addition, this Astrin-SKAP complete complex does not tolerate the low salt conditions needed for microtubule binding assays. Therefore, we worked to find a minimal truncation that would improve Astrin-SKAP complex behavior without disrupting its key features and properties required for complex assembly and its interaction with microtubules. By analyzing the cellular properties of Astrin, we were able to separate this large protein into two functional regions – a C-terminal domain that is sufficient to localize to kinetochores, but does not associate with the rest of the complex, and an N-terminal region (amino acids 1-693) that is sufficient for complex association and microtubule binding. The Astrin-SKAP complex purified with this Astrin truncation is much better behaved (salt tolerant, etc.), but we are still unable to achieve substantial amounts or concentrations of this complex. Importantly, we were able to demonstrate clear microtubule binding for this truncated Astrin-SKAP complex and demonstrate a clear synergy in the Ndc80 complex mixing experiments.

For the revised paper, we attempted to conduct a modified version of the Friese et al. experiment in which we added the Astrin-SKAP complex to assess the behavior of the Ndc80 complex. The prediction from the Friese et al. work is that the presence of the Astrin-SKAP complex would compete off the Ndc80 complex from microtubules. For these experiments, we used limiting concentrations (~30 nM) of the Ndc80 complex and added in as much Astrin-SKAP complex as was feasible to obtain. However, even when isolating the Astrin-SKAP complex from ~1 L of insect cell culture and using the entire purification for one round of microtubule binding assays, we were unable to obtain final concentrations of the Astrin-SKAP complex above ~250 nM. The data from two versions of these binding assays are shown in Author response image 1. In each case, the Astrin-SKAP complex did not compete off the Ndc80 complex. Instead, Astrin-SKAP complex modestly increased the microtubule binding activity of the Ndc80 complex, as would be predicted from the reciprocal experiments. However, based on the behavior of the Astrin-SKAP complex at these higher concentrations and variability in the purification behavior towards achieving these amounts, we have chosen not to include this data in the paper.

For the prior version of the paper, we chose to use the Ndc80 complex “Bonsai” version. This internal truncation of the Ndc80 complex removes some coiled-coil regions and a “loop” but contains the extensively defined microtubule binding activity of the complex, and can be purified and concentrated to very high levels. This construct allowed us to conduct the competition experiment with sufficient concentrations to demonstrate a clear synergy with the Astrin 1-693 complex, but to compete off an Astrin-SKAP construct (Astrin 465-693) that does not have the necessary Astrin interaction domain for synergistic binding. To conduct these competition experiments (Figure 5), we required a starting concentration of 48 µM Ndc80 Bonsai complex to achieve the final concentrations in the reaction mix. Our lab has not purified the full-length human Ndc80 complex before, and technical challenges remain for achieving this. However, we note that based on our attempts and the nature of the published literature on this complex from the Mussachio lab, it is evident that it would not be obtainable at the levels and concentrations needed for our assays.

For the revised paper, we have conducted some related experiments with the Ndc80 Broccoli complex (see Schmidt et al. 2012). This complex lacks the “roots” of the complex that mediate its kinetochore targeting, but contains the entire N-terminal regions of Nuf2 and Ndc80, including an internal Ndc80 “loop” region, thereby providing a complex highly analogous to full length Ndc80 for microtubule binding studies. Although we are unable to obtain this complex at the concentrations required for the saturation competition experiments, we were able to conduct the synergy experiments as in Figure 5. As for the prior work with the Ndc80 Bonsai complex, we found that the Ndc80 Broccoli complex substantially enhanced the apparent microtubule binding activity of the Astrin-SKAP complex. This data is included in Figure 5—figure supplement 1. However, we note that this assay has additional considerations related to the fact that the Ndc80 complex causes microtubule bundling when present at higher concentrations. Thus, although we have included these experiments for comparison, we would particularly like to highlight the competition experiments in Figure 5 for indicating how potent and specific the Ndc80-Astrin-SKAP interaction is on microtubules.

2) The authors should assay more thoroughly the requirement for synergistic binding in vivo for chromosome attachment, behaviour or segregation. This is important because Friese et al., Nat Comm. published that Ndc80 does not bind synergistically to microtubules. The authors could compare the behaviour of full length and the 456 truncation mutant, for example.

For the revised version, we have conducted substantial additional experimentation to analyze Astrin behavior in human tissue culture cells. For these experiments, we went back to the drawing board to create a robust system for generating gene replacements for Astrin. We utilized a CRISPR/Cas9 inducible knockout system to target endogenous Astrin combined with a transient BacMam expression system to introduce our large constructs. Astrin depletion following induction of these knockouts results in chromosome mis-alignment and spindle multipolarity (a hallmark of severe chromosome mis-alignment defects – see McKinley et al. 2017). We were able to rescue this phenotype by BacMam-based expression of full length Astrin in which the sequence was hardened against the CRISPR guides. In contrast, we found that the Astrin 694-1193 construct is fully defective in chromosome alignment. This mutant localizes to kinetochores, but is unable to associate with the rest of the Astrin-SKAP complex or bind to microtubules. Finally, the Astrin 465-1193 construct was able to rescue the most severe mitotic defects. The 465-1193 truncation associates with SKAP and other subunits. Thus, the most critical activity for this complex is to associate with microtubules at aligned kinetochores, whereas the deletion of the N-terminus does not result in catastrophic phenotypes. Providing this information provides an important context for the various roles of these regions, but dissecting the specific contributions of this domain will require more targeted phenotypic studies, which are beyond the scope and time feasible for this paper. For example, this mutant may show synthetic interactions with mutants that perturb the Ska1 complex, which also associates with the Ndc80 complex, but generating this double mutant represents a complex technical challenge. These additional cellular data are included in Figure 3.